# Synthesis and Characterization of PdAgNi/C Trimetallic Nanoparticles for Ethanol Electrooxidation

**DOI:** 10.3390/nano11092244

**Published:** 2021-08-30

**Authors:** Ahmed Elsheikh, James McGregor

**Affiliations:** 1Mechanical Engineering Department, South Valley University, Qena 83511, Egypt; 2Department of Chemical and Biological Engineering, University of Sheffield, Sheffield S1 3JD, UK

**Keywords:** intermetallic XRD patterns, alloy formation, ethanol electrooxidation, metal segregation, borohydride reduction, fuel cells

## Abstract

The direct use of ethanol in fuel cells presents unprecedented economic, technical, and environmental opportunities in energy conversion. However, complex challenges need to be resolved. For instance, ethanol oxidation reaction (EOR) requires breaking the rigid C–C bond and results in the generation of poisoning carbonaceous species. Therefore, new designs of the catalyst electrode are necessary. In this work, two trimetallic Pd_x_Ag_y_Ni_z_/C samples are prepared using a facile borohydride reduction route. The catalysts are characterized by X-ray diffraction (XRD), Energy-Dispersive X-ray spectroscopy (EDX), X-ray photoelectron Spectroscopy (XPS), and Transmission Electron Microscopy (TEM) and evaluated for EOR through cyclic voltammetry (CV), chronoamperometry (CA), and electrochemical impedance spectroscopy (EIS). The XRD patterns have shown a weak alloying potential between Pd, and Ag prepared through co-reduction technique. The catalysts prepared have generally shown enhanced performance compared to previously reported ones, suggesting that the applied synthesis may be suitable for catalyst mass production. Moreover, the addition of Ag and Ni has improved the Pd physiochemical properties and electrocatalytic performance towards EOR in addition to reducing cell fabrication costs. In addition to containing less Pd, The PdAgNi/C is the higher performing of the two trimetallic samples presenting a 2.7 A/mg_Pd_ oxidation current peak. The Pd_4_Ag_2_Ni_1_/C is higher performing in terms of its steady-state current density and electrochemical active surface area.

## 1. Introduction

Fuel cells are electrochemical energy conversion devices capable of converting the fuel chemical energy into electricity [1,2]. They possess multiple technical and environmental advantages considering the decaying fossil fuel resources and growing energy needs. Their operation is theoretically 100% efficient because they are not restricted by the Carnot thermal/mechanical energy conversion limitation of conventional heat engines [3]. Direct ethanol fuel cells (DEFCs) are fed with liquid ethanol instead of hydrogen. DEFCs are friendly to the environment and present high flexibility to supply diverse energy needs. As a liquid, ethanol is easy to store and transport and presents a high energy density (8 kWh/kg) [4,5,6,7]. Furthermore, it can be produced from biomass resources such as sugar cane, corn, and, more recently, from diverse agricultural and biomass wastes and residues which means it is sustainable and carbon-neutral [5]. Despite these advantages, the full commercialization of DEFCs is far from being realized. There are difficult-to-overcome challenges mainly in the form of cell fabrication costs. All low-temperature fuel cells need active noble metal (Pt) catalysts to push the fuel oxidation and oxidant reduction reactions forward. Another challenge is that most C-containing fuels generate CO–species during their oxidation. These poison catalytic sites and prevent further reaction, even if present only in low concentrations, e.g., 50 ppm [8,9]. Additionally, ethanol contains a strong C–C bond thereby favoring its incomplete oxidation to acetate instead of CO_2_ and releasing only 4 electrons instead of 12 [10,11]. Pt is very active for fuel cell electrocatalysis but is very expensive and scarce at the same time. Its expense can account for half of the cell fabrication [12]. There is, therefore, a need to find a Pt-alternative that presents comparable activity but also is more abundant to decrease the cell fabrication cost. Pd is more abundant than Pt and has shown a comparable performance for fuel cell electrocatalysis in alkaline electrolyte [13,14,15]. The fluctuating Pd and Pt prices are a complex phenomenon that involves technological, economic, and political contributions [16,17]. South Africa, Russia, Zimbabwe, Canada, and USA have Pd and Pt deposits. Prior to 1990, Pd was generally cheaper than Pt to mine and produce. However, with the advancement of Pd-based autocatalytic systems by the mid-90s, the demand and cost of Pd soared. By 2000, the price of Pd was much higher than that of Pt [18]. However, in 2007, the prices of Pt and Pd were reported $1450/ounce and $450/ounce, respectively [19]. Still, however, Pd is more attractive for automakers because it is not easy to switch back to Pt-catalytic converters. Furthermore, the diversity of Pd production by country and region is higher than that of Pt. South Africa, solely, controls Pt production while Pd mining capabilities are shared by Russia, South Africa, and North America [18]. This is an important factor due to the potential of a future shortage of Pt which is less likely in the case of Pd [20]. Another advantage of Pd over Pt in this particular EOR application is that it is more CO tolerant than Pt, however, Pt is more CO_2_ selective than Pd [21]. To achieve a high-surface-area, Pd nanoparticles are dispersed on an inert, conducting, and porous material such as carbon black. 

To further enhance the performance pf Pd and to decrease cell fabrication expenses, it is recommended to add another metal (or two) during the synthesis. The cocatalyst metal is expected to activate water and generate OH species to facilitate the oxidation of adsorbed ethoxy species. Moreover, it would modify the Pd geometry and electronic configuration and tune in its adsorption characteristics. Ni has been proven a beneficial cocatalyst when added to Pd because it can generate OH at a lower applied potential and modify the electronic structure of Pd [22,23,24,25,26,27]. The coexistence of Ni and Ni(OH)_2_ can enhance alcohol oxidation on transition metals in acidic and basic electrolytes [6,28]. Feng et al. [29] prepared unsupported porous Pd and PdNi catalysts for ethanol electrooxidation sowing that PdNi exhibits enhanced electrocatalytic performance when compared to monometallic Pd. This enhancement is ascribed to the electronic and bifunctional effects of Ni. Zhang et al. [27] prepared various-proportion Pd_x_Ni_y_/C catalysts for ethanol oxidation reaction. They found that Ni can generate oxygen species at lower applied potential, recovering Pd active sites and thus promoting ethanol oxidation, while through the microemulsion syntheis method it was possible to control PdNi particle size and make efficient contact between Pd and Ni. You et al. [30] prepared bimetallic PdAg dendrites with various composition and porous structure and evaluated them towards ethanol oxidation. Alloying between Pd and Ag shifts up the Pd d-band center leading to more tolerance for intermediates and poisons during EOR. Elsewhere, Li et al. [31] prepared PdAg nanoparticles supported on reduced graphene oxide (RGO), noting enhanced ethanol and methanol oxidation on PdAg when compared to Pd only. Oliveira et al. [32] prepared PdAg alloys and tested their efficacy for oxygen reduction reaction (ORR) and EOR. The kinetics of both reactions are promoted on PdAg as compared to Pd only. Additionally, the alloys maintained a higher selectivity for ORR in presence of ethanol. Similar conclusions on the benefits of adding Ag to Pd have been reported elsewhere [33,34].

In this work, trimetallic samples of C-supported PdAgNi nanoparticles are prepared, characterized, and evaluated for ethanol oxidation reaction (EOR) for the first time. Pd-based bimetallic nanocatalysts have been extensively investigated and reported for fuel cell electrocatalysis and EOR. Yet, a few groups have pursued the synthesis and application of C-supported trimetallic samples for similar purposes [35,36,37,38,39,40,41,42,43]. It is postulated that adding two metals is likely to provide benefits towards ethanol oxidation through altering Pd geometry, surface configuration, adsorption capacity, and coordination [35,37,38,39,44,45]. However, several factors such as the metal chemistry, synthesis method, reducing agent, and support structure may adversely affect the outcome. The ultimate objective is to design new electrodes that are cost-effective, EOR-active, CO-tolerant, and CO_2_-selective.

## 2. Materials and Methods

The applied borohydride reduction synthesis method follows [46,47,48]. Table 1 shows the added precursor quantities of Vulcan carbon (XC72R), PdCl_2_, NiCl_2_, and AgNO_3_ to prepare the monometallic and trimetallic samples. The theoretical metal loading was fixed at 12 Wt. %. The metal and carbon precursors were sonicated in a mixture of 2-propanol and water (50/50 *v*/*v*) for a few minutes. The theoretical Pd:Ag:Ni molar ratio is 57:28:15 and 34:33:33 for Pd_4_Ag_2_Ni_1_/C and PdAgNi/C, respectively. The metallic salts were purchased from Sigma-Aldrich (Gillingham, United Kingdom) and the vulcan carbon precursor from Cabot Corp (Boston, MA, USA).

KBr was added as a capping agent following the anion exchange method; the larger Br^−^ ion is capable of replacing the smaller Cl^−^ ion in the vicinity of Pd [49,50]. Consequently, Br^−^ ions can surround the recently reduced metal hindering its coalescence. The KBr/Metal atomic ratio was adjusted to 1.5. After that, the mixture was stirred for 10 min followed by adding the NaBH_4_ solution (0.5 M, 15 mL) in one portion. Subsequently, the whole mixture was vigorously stirred for 30 min. Finally, the wet powder was dried at 80 °C in vacuum oven overnight. 

X-ray diffraction (XRD) was undertaken to analyze the catalyst structure. The equipment is a Bruker D2 Phaser (Billerica, MA, USA) using Cu Kα radiation at 30 kV, 10 mA, and 12°/min scan-rate. The chemical composition of the catalysts was examined through energy-dispersive X-ray (EDX) spectroscopy conducted on a JEOL JSM 6010LA scanning electron microscope (SEM) (Akishima, Tokyo, Japan). Each catalyst surface was examined twice applying two different accelerating voltages: 10 kV and 20 kV. The different accelerating voltages change the interaction volume resulting from the electron beam-sample surface interaction. Thus, applying two different voltages enables the composition at two different depths to be studied [51]. Transmission electron microscopy (TEM) was undertaken to examine the catalyst surface morphology using a Phillips C100 microscope (Hillsboro, OR, USA) operating at 100 kV and equipped with a LaB6 filament. X-ray photoelectron spectroscopy (XPS) was applied to investigate the metal oxidation state and surface composition of the prepared electrocatalysts. A Thermo Scientific K-Alpha^+^ spectrometer (Waltham, MA, USA) equipped with an Al-X ray source (72 W) was used. The pass energy to record the data was 150 eV for survey scans and 40 eV for high-resolution scans. The survey scan step size is 1 eV, and that of the high-resolution scans is 0.1 eV. Low-energy electrons and argon ions were used to neutralize the charge. CasaXPS (Teignmouth, UK) was used to analyze the data implementing a Shirley-type background and Scofield cross-sections with an energy dependence of −0.6. 

The catalysts were evaluated for EOR via cyclic voltammetry (CV), chronoamperometry (CA), and electrochemical impedance spectroscopy (EIS). For this, a 3-electrode half-cell was used in which the working electrode is a glassy carbon electrode (GCE, Ø 3mm) coated with each catalyst ink. The reference and counter electrodes were Ag/AgCl (sat’ KCL) and Pt wire, respectively. The applied potential is converted to normal hydrogen electrode (NHE). A Gamry 600 portable Station from Gamry Instruments Inc. (Warminster, PA, USA) was used to perform the electrochemical evaluation. The catalyst ink was prepared by dispersing 5 mg of its powder in 25 µL (Nafion^®^ 117, 5%) and 2000 µL of ethanol (100%). The mixture was then sonicated for 1 h. Before drop-casting the working electrode with ink, it was polished with alumina powder (1 and 0.05 µm, respectively) to produce a mirror-like surface. Then, 25 µL of the ink was painted on GCE at 5 µL intervals.

## 3. Results

### 3.1. Physical Characterization

#### X-ray Diffraction (XRD)

Figure 1 shows the XRD patterns of Pd/C, Ni/C, Ag/C, PdAgNi/C, and Pd_4_Ag_2_Ni_1_/C. The 25° peak present in all patterns is attributed to the semi-crystalline graphitic nature of Vulcan carbon. For the Ni pattern, the metal peaks are of lower intensity than the carbon peaks as Ni is present in the form of hydroxide and oxide, indicated by peaks at 34°, 60°, and 42.5°, respectively. The Ag pattern, on the other hand, shows the (111), (200), (231), (220), and (311) reflections at 38°, 44°, 46°, 65°, and 77°, respectively resembling a face-centered cubic Ag (JCPDS card, File No. 04-0783) [52]. A similar Ag pattern has been obtained previously [31]. The Pd pattern presents (111), (200), (220), and (311) reflections located at 39.8°, 46°, 76.7°, and 82.2° indicative of Pd (JCPDS card, File No. 46-1043). Considering the XRD patterns of both trimetallic samples, it can be noted that a significant peak overlapping occurs. The high intensity peaks present at lower-angles align well with the pure Ag peak positions, whereas the lower-intensity, broader, peaks at higher-angles align with the peak positions of Pd. Monometallic Ni/C shows Ni(OH)_2_ and Ni (111) peaks at 34.5°, 60.1°, and 42.7°, respectively.

### 3.2. Energy Dispersive X-ray Spectroscopy (EDX)

Table 2 lists the elemental metal molar and weight compositional ratios of PdAgNi/C and Pd_4_Ag_2_Ni_1_/C. While the average total metal loading of Pd_4_Ag_2_Ni_1_/C as measured by EDX is close to the theoretical loading at ~12 wt. %, it is slightly higher (16 wt. %) in the case of PdAgNi/C. As EDX is a surface-weighted technique, this indicates the preferential distribution of metal in the surface region of the catalyst. Furthermore, higher Ni atomic concentrations are observed at 10 kV than at 20 kV indicating the preference of Ni to be distributed in the near-surface region. 

The PdAgNi/C surface can therefore be considered as relatively rich in Ni, while for Pd_4_Ag_2_Ni_1_/C the Ni content in the surface region is comparable to that of Ag even though the theoretical Ni quantity in the bulk sample is half of that of Ag. Figure 2 shows the elemental EDX maps of Pd, Ni, Ag, and C in both PdAgNi/C and Pd_4_Ag_2_Ni_1_/C. Examining the maps of Pd_4_Ag_2_Ni_1_/C (A–D) shows that the carbon support and metal species are evenly dispersed across the catalyst surface. 

### 3.3. Transmission Electron Microscopy (TEM)

Figure 3 shows TEM micrographs (A, C, D) of Pd/C, Pd_4_Ag_2_Ni_1_/C, and PdAgNi/C. The respective particle size distribution is, also, shown (B, D, F). From inspection of the Pd/C micrograph, it can be noted that Pd nanoparticles (average size 6.7 nm) are dispersed over larger carbon aggregates (40–60 nm). Additionally, some particle agglomeration can be noted. The particle size distribution was measured by manually selecting the metal particles through use of ImageJ software. The average particle size and standard deviation of Pd/C are 6.7 and 5 nm, respectively. The high standard deviation value is suggestive of the high variation in particle size as a consequence of high particle agglomeration. In contrast, the average particle size of Pd_4_Ag_2_Ni_1_/C is 5.6 nm and the standard deviation to 2.7 nm. This is possibly due to the contradicting forces that affect the particle aggregation during both nucleation and growth. It is extensively reported that preparing multi-metallic samples results in different attraction and repulsion forces and, therefore, smaller particle sizes [22,24,41,53,54,55,56]. The same trend is noted in PdAgNi, which has a smaller total quantity of Pd; in this case the average particle size and standard deviations are 4.4 nm and 3 nm, respectively.

### 3.4. X-ray Photoelectron Spectroscopy (XPS)

XPS analysis was undertaken to investigate the metal oxidation state and surface composition of the samples. Figure 4 shows the XPS full survey scans of Pd/C, PdAgNi/C, and Pd_4_Ag_2_Ni_1_/C. The C1s, Pd 3d, Ag 3d, Pd 3p (overlapping with O 1s), and Ni 2p are located approximately at 284, 335, 370, 532, and 856 eV, respectively. 

Figure 5 shows the detailed elemental peaks of Pd 3d, Ag 3d, and Ni 2p in Pd/C, Pd_4_Ag_2_Ni_1_/C, and PdAgNi/C. The Pd 3d of Pd/C (A) is deconvoluted into a high-energy band (340.4 eV) of Pd 3d_3/2_and a low-energy band (335.48 eV) of Pd 3d_5/2_. Additionally, the PdO peaks are visible around 342 eV and 337 eV. Table 3 lists the XPS surface concentrations of C, O, Pd, Ag, and Ni for the three samples. The XPS surface concentration Pd of Pd/C is 1.63 At. % and 0.45% for Pd^0^ and Pd^2+^, respectively. The presence of Pd oxide is indicative of Pd air instability which could be improved by adding another metal. PdAgNi/C also presents a significant fraction of Pd^2+^—16% of the total Pd present. Pd_4_Ag_2_Ni_1_/C however, with a lower overall Pd content, presents only metallic Pd^0^, as shown in Figure 5D. The Ag 3d double-peak is located at 374 and 368 eV for 3d_3/2_ and 3d_5/2_ (Figure 5E), respectively. The Ni 2p peaks of Ni^0^ and Ni^+2^ of Pd_4_Ag_2_Ni_1_/C are present at 852, 856, and 862 eV (Figure 5D), respectively, with no visible satellite peaks. The Ni XPS surface concentration of that sample is 0.05 at. % (for metallic Ni) and 0.73 at. % (for Ni(OH)_2_). 

### 3.5. Electrochemical Evaluation

Figure 6 shows the cyclic voltammograms of Pd/C, PdAgNi/C, and Pd_4_Ag_2_Ni_1_/C performed in 1M KOH at 50 mV/s. It is noteworthy that the thinnest profile is observed for Pd/C. This is indicative of a thinner double-layer, and consequently, faster charging/discharging behavior for Pd/C [42]. As Pd is known to absorb hydrogen within its bulk, the H_abs/ads_ peak in the forward scan at ~−600 mV is not present. This peak is, however, more pronounced on both trimetallic samples, which may be ascribed to the presence of Ag. Upon increasing the potential, the OH adsorption peak between −400 and −200 mV is more apparent for monometallic Pd/C than for the trimetallic samples. OH adsorption can be considered as the onset of surface oxidation, however oxidation of Pd commences at approximately 0.0 mV vs. NHE. Ag surface oxidation is noted at ~470 mV followed by Ni oxidation up to the scan end at 600 mV. Both Ni and Ag oxidation peaks are more pronounced for PdAgNi/C than for Pd_4_Ag_2_Ni_1_/C as the former contains higher quantities of Ni and Ag. In the reverse scan, distinctive reduction peaks for Ni, Ag, and Pd appear at 450, 220, and −200 mV, respectively. A small shoulder on the Pd reduction peak is noted for Pd_4_Ag_2_Ni_1_/C and Pd/C which is not clear for PdAgNi/C. This is likely to be reflective of the reduction of different Pd oxides. 

Figure 7 shows the cyclic voltammograms of the three catalysts in 1M KOH+EtOH at 50 mV. The addition of ethanol suppresses the H_ands/abs_ peak as reported elsewhere [55,57]. With the start of OH adsorption, adsorbed ethoxy species undergo oxidation and are removed from Pd sites, thereby making these sites available for further fuel oxidation. As the adsorbed OH increases with increasing potential, the free Pd sites increase, and an increasing current is drawn. While the onset oxidation potential in case of Pd/C is −390 mV, it is shifted to −500 mV in case of PdAgNi/C and Pd_4_Ag_2_Ni_1_/C which suggests a reduction in the activation barrier against ethanol oxidation on the surface of both trimetallic samples. A similar shift in potential is noted for the oxidation peak on PdAgNi sample compared to Pd. The peak current density in case of PdAgNi/C (2700 mA/m_gPd_) is higher than Pd_4_Ag_2_Ni_1_/C (2300 mA/m_gPd_) and the smallest peak current density is of Pd/C (1.8 A/m_gPd_).

When the catalyst surface can no longer adsorb OH, the surface Pd oxides thereby decreasing the number of Pd active sites as the potential increases. This ultimately decreases the drawn current up to the end of the forward scan. It is noteworthy that around the forward current peak, the reactants are depleted much faster than at the beginning of the scan [8]. A shoulder peak is noted on PdAgNi/C at 200 mV which may be suggestive of the oxidation of ethoxy intermediates on Pd. At the end of the PdAgNi forward scan, a sharp current rise is noted due to the Ni oxidation that also appears in the voltammogram conducted in the absence of ethanol. The sharp peak in the inverse scan is due to the removal of the incompletely oxidized intermediates from EOR and recovery of Pd active sites. It is noteworthy that the forward sweep current peak is higher than the reverse one on all catalysts which is a positive indication of the ability of the catalyst to tolerate poisoning species [58,59]. Therefore, CO-like species are less likely to block further EOR on the catalysts developed in the present work. 

Figure 8 shows the chronoamperometric scans of Pd/C, Pd_4_Ag_2_Ni_1_/C, and PdAgNi/C performed at fixed potentials of −300 mV and +100 mV vs. NHE. The former was chosen because it is in the middle of the window in which OH was adsorbed causing the oxidation and removal of ethoxy species from Pd surface. The latter was chosen because it is greater than the potential at which the Pd surface oxidation starts. These data show that the CA current density of PdAgNi/C sample—though higher than that of Pd_4_Ag_2_Ni_1_/C at the two CA steps—decays much faster at the higher-potential (+0.1 V) to the extent that it drops below Pd_4_Ag_2_Ni_1_/C towards the end of the CA duration. According to EDX and XPS, the surface of PdAgNi/C is Ni-rich and correspondingly deficient in Pd and Ag. PdAgNi/C, therefore, presents fewer accessible Pd surface sites for EOR than Pd/C or Pd_4_Ag_2_Ni_1_/C. During the high-potential CA scan, the number of available Pd sites further decreases through site poisoning via the strong adsorption of EOR intermediates. The PdAgNi voltammograms (Figure 6 and Figure 7) show that OH adsorption does not continue at +0.1 V. Therefore, at high potential, PdAgNi/C is susceptible to poisoning by carbonaceous species from EOR. In contrast, Pd_4_Ag_2_Ni_1_/C generates a steadier CA current density than PdAgNi/C at +0.1 V; which is ascribed to the higher concentration of Pd in the surface region for the former, providing a greater number of available Pd sites and hence greater resistance to poisoning. Pd/C achieves an intermediate current decay between PdAgNi/C and Pd_4_A_g_2Ni_1_/C which could be explained by two factors: the strong adsorption of CO-like species and the abundance of Pd surface sites. 

Figure 9 shows the EIS spectra of Pd_4_Ag_2_Ni_1_/C, and PdAgNi/C recorded at −0.3 V, 0.0 V, and +0.3 V vs. NHE between 10,000 Hz and 0.1 Hz while the voltage amplitude is 5 mV. Each experiment was preceded with a 10-min potentiostatic scan to compensate for the current perpetuation. EIS spectra at −0.3 V and 0.0 V represent, to some extent, a semi-circle due to the interchangeable interaction of the double-layer capacitance through the electrolyte-electrode interface and the charge-transfer resistance. However, at +0.3 V, the effect of charge-transfer resistance is such that with further decreasing frequency, the arc continues to rise vertically. This is a consequence of more than one factor. For instance, increasing the applied potential affects the charge-transfer resistance more significantly than the double-layer capacitance. Moreover, at +0.3 V no further EOR occurs due to the Pd surface oxidation and current decrease (Figure 7). It is noteworthy the arc size at −0.3 V and 0.0 V of Pd_4_Ag_2_Ni_1_/C is slightly smaller than their counterparts of PdAgNi/C. 

Gamry Echem Analyst software was used to construct an electrical model (Figure 10) that represents the physical phenomena contributing to the impedance. R_s_ represents the solution resistance measured at the intersection point with the *x*-axis (−Z’’= 0, frequency = 10 kHz) and for both catalysts, it is approximately 14 Ω and does not change with the applied potential. The constant-phase element (Ø) represents the double-layer capacitance effect due to opposite-charge accumulation at the electrode-electrolyte interface. R_ct_ represents the charge-transfer resistance that reflects on the specific activity of each catalyst. R_ct_ values of Pd_4_Ag_2_Ni_1_/C are estimated to be 150 Ω and 183 Ω at −0.3 V and 0.0 V vs. NHE, respectively. Larger values of 285 Ω and 290 Ω—at the same respective potentials—are obtained with PdAgNi/C. This further demonstrates the improved performance of Pd_4_Ag_2_Ni_1_/C towards EOR when compared to PdAgNi/C. It is difficult to estimate the R_ct_ value at +0.3 V due to the complex phenomena encountered and missing EOR at such high potential and several EIS models were inaccurate enough to estimate reasonable R_ct_ values at that potential. 

## 4. Discussion

The XRD peak separation of Pd and Ag (Figure 1) implies that there is substantial separation of Pd and Ag in the nanoparticles and therefore that it is less likely a nanoalloy has been formed. Separate Ag and Pd peaks were also reported by other researchers [31,60], while other groups have found singular peaks that resemble both Pd and Ag [30,39,61]. Olivera et al. [32] annealed electrolessly deposited Pd and Ag films on steel discs to produce Pd-Ag alloy films. Before annealing, distinctive Pd and Ag phases were detected by XRD, with a single peak observed after annealing. The Ni(OH)_2_ individual peak disappearance in the XRD patterns of both trimetallic samples is suggestive of a high degree of mixing between Pd and Ni whose atoms could be assumed incorporated into the Pd lattice. The separate peaks of Ni(OH)_2_ are commonly seen when synthesizing bimetallic and trimetallic Ni-containing catalysts [22,29,53,54,55,62,63]. The surface metal loading of PdAgNi/C (16 wt. %), as measured by ESX, is 4 wt. % higher than the nominal bulk loading (Table 1), with Ni segregating to the surface. The opposite trend is noted for Ag which segregates into the catalyst core. The Pd: Ag: Ni molar ratio at 10 kV is 4:1:2 while it is 4:1.6:1 at 20 kV which shows the Pd concentration does not change significantly with analysis depth, but the Ag and Ni ones do. A similar observation is noted for PdAgNi/C whose Pd:Ag:Ni at 10 kV is 2:1:4 while it is 1.75:1:2.25 at 20 kV. 

According to the XPS measurements (Table 3), Ag is present exclusively in the metallic state with a concentration of 0.4 at. % and 0.32 at. % for Pd_4_Ag_2_Ni_1_/C and PdAgNi/C, respectively. This is despite the theoretical Ag loading being lower in Pd_4_Ag_2_Ni_1_/C (Table 1). It is noteworthy that XPS atomic Pd:Ag:Ni ratio is 4:1:2 while the ratio from surface-weighted EDX spectroscopy measurements (Table 2) is 4.4:1.7:1. The higher Ni XPS proportion signifies its segregation into the top surface unlike Ag. In PdAgNi/C, which has a lower fraction of Pd than Pd_4_Ag_2_Ni_1_/C, the Pd air stability is lost and 15% of Pd exists in an oxidized form (Figure 5E) Figure 5G shows the Ni 2p peak of PdAgNi/C and two satellite peaks which are potentially due to the multiple Ni excitations. No metallic Ni was detected for this sample and the Ni surface concentration equals 2.01 at.%. Therefore, the XPS-derived Pd:Ag:Ni molar ratio is 2.7:1:6.7 while the EDX spectroscopy derived ratio (Table 2) is 1.7:2.3:1. Once again this is probably due to the Ni tendency to segregate into the surface and Ag and little Pd tendency to segregate into the core. In case of trimetallic samples, the Pd 3d peak is shifted 0.05 eV (Figure 5C) to higher binding energy than Pd/C, which may be suggestive of an electron loss from Pd to either Ag or Ni. Figure 5C shows the valence band of Pd/C, Pd_4_Ag_2_Ni_1_/C, and PdAgNi/C which indicates the *d*-band center of the two trimetallic samples are shifted approximately 1 eV from the Fermi level compared to their monometallic counterpart. Note that the impact of particle size on *d*-band center can be neglected as the particle size is ~5 nm and instead shifts in the *d*-band center are the consequence of charge transfer from Pd to other metals [35,64]. 

As H_abs/des_ peak is missing in Figure 6, it is more appropriate to estimate electrochemical active surface area (ECSA) using the PdO reduction peak in the reverse scan. The PdO reduction area in the reverse scan is the smallest in case of Pd/C while higher reduction currents are obtained with PdAgNi/C and Pd_4_Ag_2_Ni_1_/C. The PdO reduction peak is used as a measure of the Pd fraction exposed on the surface, and therefore, available to undertake redox reactions. The electrochemical active surface area (ECSA, cm^2^/mg) is calculated according to [26]: $$ECSA=\frac{Q}{0.43\times \left[Pd\right]}$$
where *Q* is the charge extracted from PdO reduction peaks in mC, 0.43 mC/cm^2^ is the monolayer reduction charge of Pd, and [Pd] is the weight of Pd on GCE in mg. This indicates ECSA is highest on Pd_4_Ag_2_Ni_1_/C followed by PdAgNi/C and finally Pd/C (Table 4). This could be explained by the smaller particle size of both trimetallic samples compared to the monometallic one according to TEM (Figure 3). The three catalysts have very significantly larger ECSAs compared to the other materials listed in Table 4. This is indicative of the high Pd surface fraction; although Ni has a higher tendency to segregate into the surface as shown by EDX and XPS analyses. The low overall metal loading (12 wt. %) decreases the overall metal surface energies and aggregation potential leading to a thicker and rougher catalyst layer, while the KBr capping works against agglomeration by surrounding the individual nanoparticles during synthesis with Br^−1^ ions. The ethanol-enhanced CV performance on both trimetallic samples is a reflection of the Ag and Ni effects. This indicates that more EOR Pd active sites exist on the trimetallic sample surfaces than on Pd/C. A similar finding was previously reported regarding the activity of PdAgNi towards formic acid oxidation [39]. The lower oxidation onset potentials and higher forward current peak are advantageous for the three samples in this work compared to many other previously published catalysts [58,65,66,67]. Table 4 compares the EOR onset oxidation current density and current peak obtained with this work catalysts and previously published ones. The reverse current peak of Pd/C, Pd_4_Ag_2_Ni_1_/C, and PdAgNi/C are smaller than their forward counterparts. This is indicative of a good Pd tolerance towards the poisoning species. When o_f_/j_b_ is >1, it implies less incompletely oxidized ethoxy intermediates on Pd surface which suggests the catalyst is more likely to recover its surface-active sites and proceed with further EOR. When it is <1, it suggests there are more incompletely-oxidized carbonaceous species on the catalyst surface that are strongly bound to active sites. Adding Ni to the Pd-based catalyst has, in some previous works, resulted in j_f_/j_b_ decreasing below 1 [29,43], although other works observe the opposite trend [42,54,68]. 

It should be noted the chronoamperometric currents drawn on trimetallic catalysts (Figure 8) are substantially higher than that on Pd/C which suggests a higher catalyst capability to recover the Pd active sites by generating OH species which indicates the addition of Ag and Ni is beneficial for EOR catalytic performance. The catalyst containing the least Pd (PdAgNi/C) draws the highest current density (−0.3 V). This implies that this surface is highly active for the generation of oxygen species, facilitating the removal of adsorbed ethoxy species. Ni surface segregation and Ag core segregation, as determined through EDX spectroscopy and XPS, may in part explain the enhanced V-*j* behavior of PdAgNi/C. A somewhat similar behavior is noted for the other trimetallic sample. Note, however, that a very high level of OH adsorption on the catalyst surface would negatively impact the overall catalyst performance as many of the EOR-active surface sites would be occupied by OH. 

The enhanced the performance of the trimetallic catalysts in the present work is likely due to Ni and Ag generating OH species resulting in the larger ECSA for both samples as shown in Figure 6. The current density drawn on three catalysts at +0.1 V is higher than at −0.3 V due to the increased activation voltage and OH adsorption. It is also noteworthy that at −0.3 V vs. NHE, the three catalysts exhibit similar behavior in terms of the current decays as the scan proceeds. However, at −0.1 V vs. NHE, Pd_4_Ag_2_Ni_1_/C shows better stability (stable current decay) than PdAgNi/C even though the former draws a higher current for the majority of the 30-min scan. Although the initial current decay at −0.3 V is similar on both Pd_4_Ag_2_Ni_1_/C and PdAgNi/C, at +0.1 V the current decay rate is higher for PdAgNi/C than for Pd_4_Ag_2_Ni_1_/C and the former draws a higher current at the scan end. Several factors likely contribute to this enhanced performance upon adding both Ag and Ni. For instance, the activation barrier of EOR on both trimetallic samples less than that on Pd/C due to the reduction of the bandgap energy, indicated by the smaller activation overvoltage [8]. Also, the large number of active sites verified by the smaller TEM particle size in the case of both trimetallic samples, will play a positive role. Another factor that may contribute to the observed higher ESCA values as compared to previous studies is that the experiment was performed with the electrolyte being stirred. This may help to ensure the uniform distribution of reactants/products and reduce the effects of mass transport and diffusion.

Compared to many reported synthesis methods, the currently applied method is quicker and easier because it is a room-temperature one-pot synthesis. Furthermore, it was concluded in a short time (30 min). Many of the previous synthesis protocols include use of high temperature and/or longer synthesis times [29,42,43,54,68]. Also, the smaller metal loading in the present work is potentially beneficial for the physical and electrochemical characteristics of the supported catalyst. Additionally, it is noted that Ni tends to segregate into the surface while Ag tends to segregate into the bulk of trimetallic particles. Furthermore, the trimetallic catalysts are cost-effective than those consisting only of platinum-group metals. The collective characterization and evaluation results suggest that Pd_4_Ag_2_Ni_1_/C is the best performing catalyst and that adding a significant Ni quantity (>35 At. %) would adversely impact the catalyst physical and electrocatalytic performance

## 5. Conclusions

C-supported PdAgNi intermetallic catalysts have been successfully prepared through a straight-forward chemical synthesis method suitable for up-scaling for the first time. Adding small quantity of Ni in trimetallic catalysts has a beneficial impact but increasing the Ni content can harm the EOR performance. A relatively low metal loading (12 Wt. %) and adding KBr as a capping agent are recommended to produce a well-dispersed catalyst with a high electrochemical surface area (ECSA). However, the followed protocol does not produce a highly order alloy structure of Pd and Ag as indicated by XRD. Although the Pd_4_Ag_2_Ni_1_/C surpassed the PdAgNi/C in terms of ECSA and ethanol oxidation potential, the latter outperformed it in terms of the oxidation current peak and steady chronoamperometric current. Adding Ag and Ni to Pd not only improves its catalytic performance towards EOR, but considerably decreases the catalyst preparation cost. While Ag tends to segregate to the bulk of the PdAgNi nanoparticle, Ni tends to segregate to its outer surface layers. Pd_4_Ag_2_Ni_1_/C combines both high activity and stability towards EOR. 

## Figures and Tables

**Figure 1 nanomaterials-11-02244-f001:**
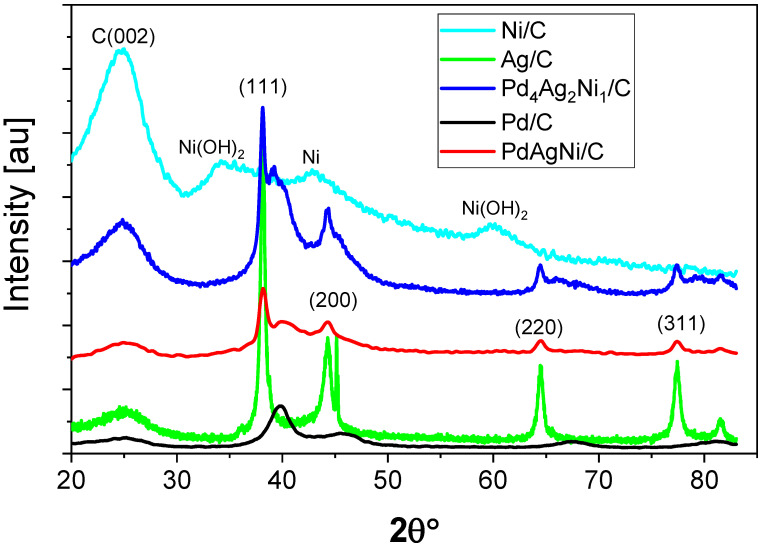
X-ray diffraction patterns of Pd/C, Ag/C, Ni/C, PdAgNi/C, and Pd_4_Ag_2_Ni_1_/C.

**Figure 2 nanomaterials-11-02244-f002:**
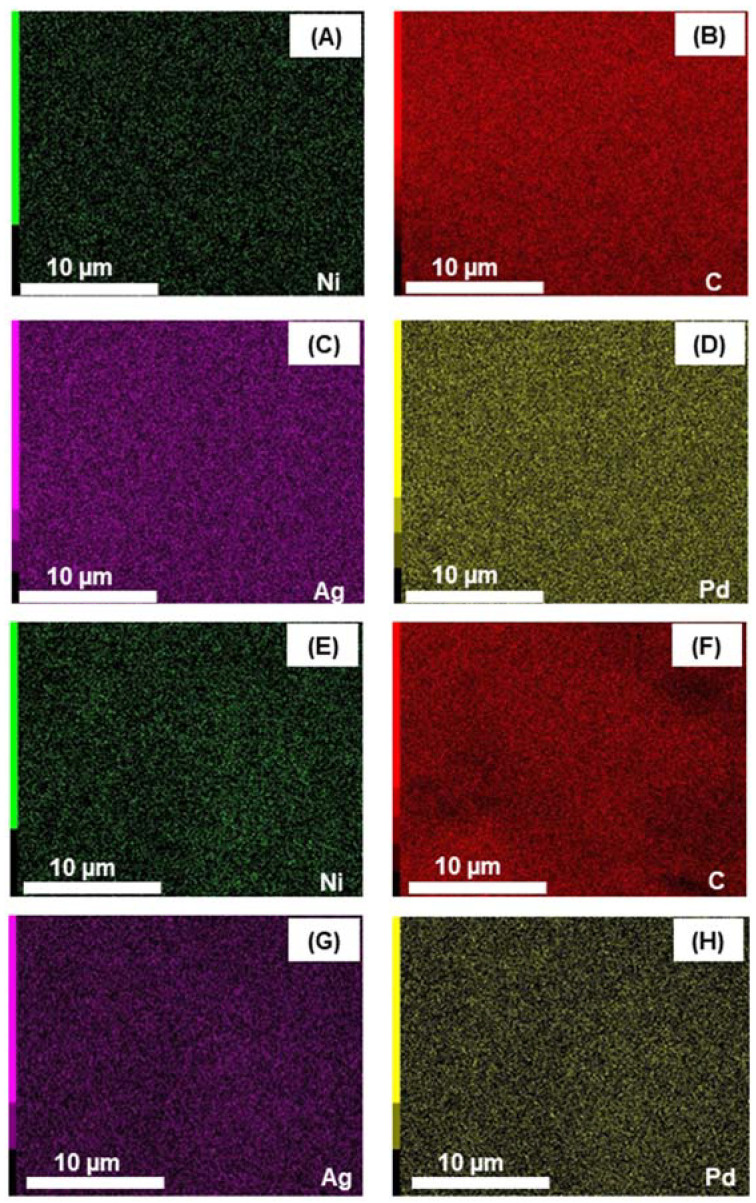
EDS elemental maps of Pd_4_Ag_2_Ni_1_/C (**A**–**D**) and PdAgNi/C (**E**–**H**) taken at 10 kV.

**Figure 3 nanomaterials-11-02244-f003:**
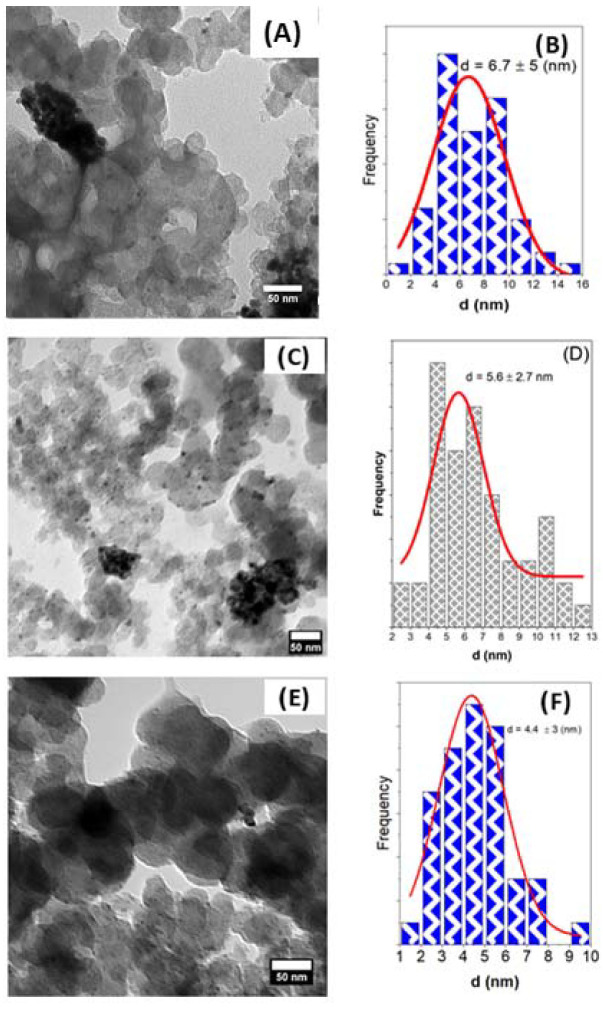
TEM Micrographs and the respective particle size distribution of Pd/C (**A**,**B**), Pd_4_Ag_2_Ni_1_/C (**C**,**D**), and PdAgNi/C (**E**,**F**).

**Figure 4 nanomaterials-11-02244-f004:**
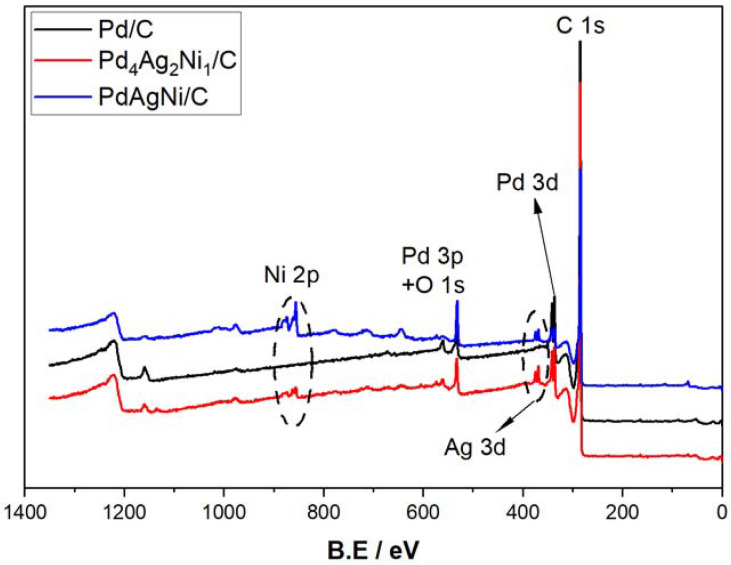
XPS full surveys of Pd/C, Pd_4_Ag_2_Ni_1_/C, and PdAgNi/C.

**Figure 5 nanomaterials-11-02244-f005:**
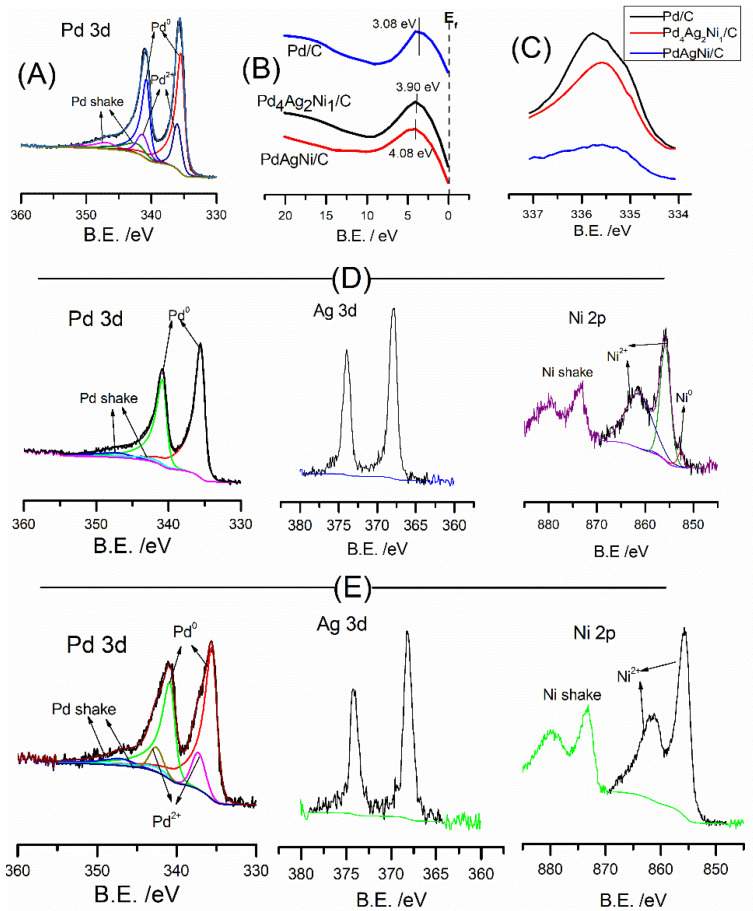
XPS elemental peaks of Pd 3d in in Pd/C (**A**), valence band spectra (**B**) and Pd 3d_5/2_ (**C**) of Pd/C, Pd_4_Ag_2_Ni_1_/C, and PdAgNi/C, Pd 3d, Ag 3d, Ni 2p, of Pd_4_Ag_2_Ni_1_/C (**D**) and PdAgNi/C (**E**).

**Figure 6 nanomaterials-11-02244-f006:**
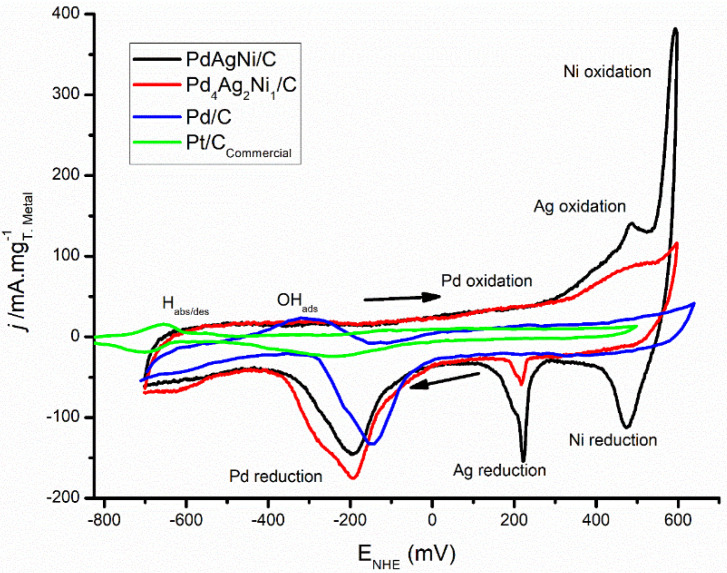
CV voltammograms of Pd/C, Pd_4_Ag_2_Ni_1_/C, and PdAgNi/C in 1M KOH at 50 mV/s.

**Figure 7 nanomaterials-11-02244-f007:**
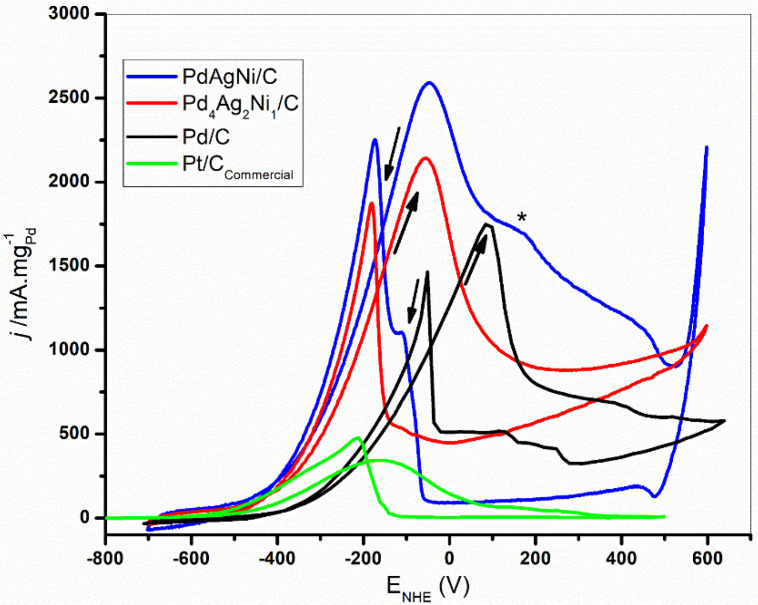
CV Voltammograms of Pd/C, Pd_4_Ag_2_Ni_1_/C, and Pd/C in 1 M KOH+EtOH at 50 mV/s.

**Figure 8 nanomaterials-11-02244-f008:**
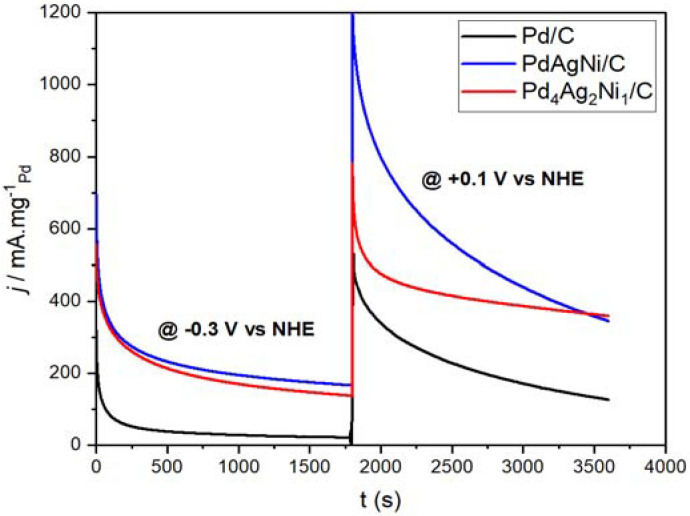
Chronoamperometric (CA) scans of Pd/C, Pd_4_Ag_2_Ni_1_/C, PdAgNi/C at −0.3 V and +0.1 V vs. NHE.

**Figure 9 nanomaterials-11-02244-f009:**
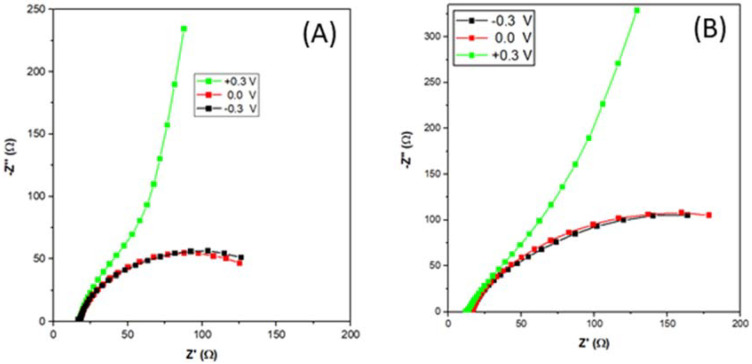
Electrochemical impedance spectroscopy (EIS) of Pd_4_Ag_2_Ni_1_/C (**A**) and PdAgNi/C (**B**) at −0.3 V, 0.0 V, and 0.3 V vs. NHE.

**Figure 10 nanomaterials-11-02244-f010:**
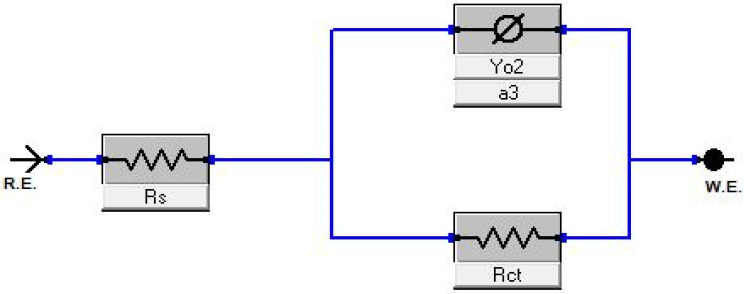
EIS electrical model representative of the physical quantities in a half-cell 3-electrode.

**Table 1 nanomaterials-11-02244-t001:** Stochiometric added quantities of C, PdCl_2_, AgNO_3_, NiCl_2_ to synthesize Pd/C. PdAgNi/C, and Pd_4_Ag_2_Ni_1_/C.

Catalyst	C (mg)	PdCl_2_ (mg)	NiCl_2_ (mg)	AgNO_3_ (mg)	Metal Wt. %
Pd/C	132	30			12
PdAgNi/C	132	11.7	8.5	11.2	12
Pd_4_Ag_2_Ni_1_/C	132	18.3	3.4	8.8	12
Ag/C	132		40		12
Ni/C	132			28.3	12

**Table 2 nanomaterials-11-02244-t002:** EDX elemental metal composition of PdAgNi/C and Pd_4_Ag_2_Ni_1_/C detected at 20 kV and 10 kV voltages.

Catalyst	Acc. Voltage	Pd		Ni		Ag		Tot. Metal Wt. %
		Wt. % *	At. % **	Wt. %	At. %	Wt. %	At. %	
Pd_4_Ag_2_Ni_1_/C	10 kV	7.89	1.02	1.64	0.38	1.98	0.25	11.51
	20 kV	8.28	1.09	1.10	0.26	3.26	0.42	12.46
PdAgNi/C	10 kV	6.47	0.88	7.68	1.90	3.39	0.46	17.54
	20 kV	6.92	0.94	5.04	1.24	4.10	0.55	16.06

* wt. %: weight percentage %; ** at. %: atomic percentage %.

**Table 3 nanomaterials-11-02244-t003:** XPS surface atomic concentration of C, O, Pd, Ag, and Ni in Pd/C, Pd_4_Ag_2_Ni_1_/C, and PdAgNi/C.

Catalyst	C At. %	O at. %	Pd at. %		Ag at. %	Ni at. %		Pd:Ag:Ni Ratio
			Pd^0^	Pd^2+^		Ni^0^	Ni^2+^	
Pd/C	96.13	1.78	1.63	0.45				
Pd_4_Ag_2_Ni_1_/C	93.91	3.14	1.59	0	0.4	0.05	0.73	4:1:2
PdAgNi/C	92.15	4.69	0.64	0.12	0.32	0	2.01	2.7: 1:6.7

**Table 4 nanomaterials-11-02244-t004:** Comparison of the ECSA (cm^2^/mg), E_onset_ (mV) and *j*_p_ (A/mg_Pd_) in 1M EtOH of this work and previously published catalysts.

Catalyst	ECSA (cm^2^/mg)	E_onset_ (mV) vs. NHE	*j*_p_ (A/mg_Pd_)	Ref.
Pd/C	1350	−390	1.8	This work
PdAgNi/C	1500	−500	2.7	
Pd_4_Ag_2_Ni_1_/C	1618	−500	2.3	
Pd/C	549	−150	0.5	[54]
Pd_83_Ni_17_	375	−250	1.1	
PdNi	209	−260	1.45	[29]
Pd	135	−209	0.8	
Pd_1_Au_1_/C	1320	−260	12	[68]
Pd/C		−260	0.75	
Pd_2_Sn_2_Ag_1_/C	243	−360	0.8	[42]
Pd_2_Sn_2_Ni_1_/C	209	−330	0.4	
Pd_2_Sn_2_Co_1_/C	212	−300	0.35	
PdAgCu	506	−360	4.56	[43]
Pd/C		−350	1.47	[21]
Pt/C		−400	0.65	

## Data Availability

Not applicable.

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
