# Peer review of "Synthesis and Characterization of PdAgNi/C Trimetallic Nanoparticles for Ethanol Electrooxidation"

_nanomaterials, 2021, doi:10.3390/nano11092244_

Round 1
Reviewer 1 Report
This manuscript reports a room temperature one-pot synthesis of PdxAgyNiz/C trimetallic nanoparticles. They synthesize C-supported PdxAgyNiz catalysts through a quick borohydride reduction route and characterize them. In the as-produced PdxAgyNiz/C catalysts, the addition of Ag and Ni contributes to the enhancement of Pd physiochemical properties and electrocatalytic performance towards EOR with a high electrochemical surface area (ECSA) as well as decreasing the catalyst preparation cost. The straight-forward chemical catalyst synthesis method reported in this manuscript is green and novel and the content of the manuscript is relatively complete, but there are still some disputes:
- It is known to us that the commercial Pd/C catalyst has good performance towards EOR at present, it is necessary for authors to compare the prepared catalysts with commercial Pd/C catalyst when characterize their catalytic performance.
- Why XPS and EDX results show that there is a lot of Ni content but the XRD pattern does not show it clearly?
- On page 8, the TEM images are the catalysts C-supported, but we can’t clearly distinguish the nanoparticles and there are particles aggregation, are there some HRTEM images about the nanoparticles?
- The XPS results show that there’re Ni and Ni2+ in PdxAgyNiz/C catalysts, Which one is responsible for the catalytic performance of the catalyst towards EOR?
- Are the three precursors added at the same time when preparing the PdxAgyNiz/C catalysts? What is the result of adding these precursors at different time?
- The XRD patterns don’t show a highly order alloy structure of Pd and Ag, what is the structure of PdxAgyNiz/C? Alloy or core-shell structure?
- Authors are suggested to determine the load of Pb by ICP-OES, which is more accurate than EDX.
- Authors are advised to check the manuscript to correct the possible spelling errors and change some indistinct images, like Fig.5 on page 10.
Reviewer 2 Report
The authors proposed to generate trimetallic nanoparticles as the catalyst for ethanol electro-oxidation. Pd has long been used for the catalyst of ethanol oxidation. The addition of Ag and Ni was proposed to improve the physicochemical properties and electrocatalytic performance of Pd. Although data was presented in this manuscript to show that the electrocatalytic performance and physicochemical properties of the developed trimetallic nanoparticle-based catalysts, several major issues are needed to be clarified and discussed before it can be accepted by this journal.
Specific comments:
1) The full term of “ORR” should be given at place, where it first appears.
2) Many typos and grammatical errors were found in the manuscript. Therefore, the proofread and editing are required for this manuscript.
3) What type of Vulcan carbon was used in this manuscript? How was the catalyst ink prepared? The details about the catalyst preparation and experimental setup should be described in the Materials and Methods.
4) It is unclear what the authors tried to say in the sentence in lines 170-173. What is the “ones” representing in this sentence?
5) The wt% and At% used in Tables should be defined either in the Table legend or in the main text. Why the observation of the Ni concentrations at 10 kV and at 20 kV can be used to explain the Ni surface segregation tendency? The authors should explain and discuss this issue in the manuscript.
6) The scale of the EDS elemental maps in Figure 2 is 10-micrometer, while the sizes of the carbon aggregates, and monometallic/trimetallic nanoparticles are about 50-60 nm and 3-12 nm, respectively (Figure 3). Therefore, it is a question what makes the authors conclude that the metal species are “uniformly distributed and homogeneously mixed (lines 195-196) from Figure 2. The observation of the uneven distribution of nanoparticles on carbons in Figure 3A, 3C and 3F also support this argument. Therefore, the authors should clarify this issue.
7) In line 237, the authors mentioned that the surface of Pd/C contains not only Pd but also Pdo and Pd2+. The corresponding concentrations for Pd but also Pdo /Pd2+ species were also determined. In addition, the authors mentioned that the Pd oxides may be eliminated by adding Ag and Ni in the trimetallic nanoparticles. However, these results were not presented and shown in Table 3. The surface concentrations of Pd but also Pdo /Pd2+ species on all three metallic materials should be indicated in Table 3.
8) The quality of Figures 5G, 5H, 5I needs to be greatly improved.
9) The poisonous effect of carbonaceous intermediates generated in the process of ethanol oxidation is an important issue for the performance of the developed trimetallic catalysts. This can be evaluated by the ratio of forward peak current to the backward peak current in Figure 7. The discussion about this issue has been discussed in several publications (Sawangphruk et al. J. Mater. Chem. A 2013, 1, 1030–1034; Pei et al., Catalysts 2021, 11, 248). An explain and discussion about the poisonous species resistance of the trimetallic nanomaterilas during the ethanol oxidation in the revised manuscript.
10) In Figure 9 the panels (A) and (B) were not indicated in the figure legend.
Round 2
Reviewer 2 Report
The manuscript has been revised as suggested and now is acceptable for publication.